



# Mapping 24 woody plant species phenology and ground forests phenology over China from 1951-2020

Mengyao Zhu[1], Junhu Dai[1,2,3], Huanjiong Wang[1], Juha M. Alatalo[4], Wei Liu[1,2], Yulong Hao[1,2], Quansheng Ge[1,2]

[1]Key Laboratory of Land Surface Pattern and Simulation, Institute of Geographic Sciences and Natural Resources Research, Chinese Academy of Sciences, Beijing, 100101, China
[2]College of Resources and Environment, University of Chinese Academy of Sciences, Beijing, 101408, China
[3]China-Pakistan Joint Research Center on Earth Sciences, CAS-HEC, Islamabad, 45320, Pakistan
[4]Environmental Science Centre, Qatar University, Doha, 2713, Qatar

*Correspondence to*: Junhu Dai (daijh@igsnrr.ac.cn); Quansheng Ge (geqs@igsnrr.ac.cn)

**Abstract.** Plant phenology refers to the cyclic plant growth events, and is one of the most important indicators of climate change. Integration of plant phenology information is of great significance for understanding the response of ecosystems to global change and simulating the material and energy balance of terrestrial ecosystems. Based on 24552 in-situ phenology observation records of 24 typical woody plants from the Chinese Phenology Observation Network (CPON), we map the species phenology (SP) and ground phenology (GP) of forests over China from 1951-2020, with a spatial resolution of 0.1° and a temporal resolution of 1 day. A model-based upscaling method was used to generate SP maps from in-situ SP observations, and then weighted average and quantile methods were used to generate GP maps from SP maps. The validation shows that the SP maps of 24 woody plants are largely consistent with the in-situ observations, with an average error of 6.9 days in spring and 10.8 days in autumn. The GP maps of forests have good agreement with the existing Land Surface Phenology (LSP) products derived by remote sensing data, particularly in deciduous forests, with an average difference of 8.8 days in spring and 15.1 days in autumn. The dataset provides an independent and reliable phenology data source on a long-time scale of 70 years in China, and contributes to more comprehensive research on plant phenology and climate change at regional and national scales. The dataset can be accessed at https://doi.org/10.57760/sciencedb.07995 (Zhu et al., 2023).

## 1 Introduction

Plant phenology refers to plant cyclic growth and development events, which are formed by adaptation to seasonal changes in climate and environmental conditions (Lieth, 1974; Schwartz, 2003). These phenological events include critical stages such as budburst, leaf unfolding, flowering, leaf coloring, and defoliation. As a highly sensitive biological indicator of climate change (Richardson et al., 2013), plant phenology is not only important for comprehending ecosystem responses to global change (Inouye, 2022; Menzel et al., 2020), but also a significant factor in simulating material and energy balance of terrestrial ecosystems (Keenan et al., 2014; Wang et al., 2020b). To be helpful for biological monitoring and predictions,



long-term, dependable plant phenology data on a global scale are greatly desired by related scientific research personnel.
Presently, such data can be procured from diverse sources (Piao et al., 2019; Tang et al., 2016), including manual in-situ
observations (Schwartz et al., 2012; Templ et al., 2018), satellite remote sensing (Bolton et al., 2020; Dixon et al., 2021), and
tower-based digital cameras (Nasahara and Nagai, 2015; Richardson et al., 2018), etc. Nevertheless, integrating large-scale
and long-term plant phenology information continues to pose a formidable challenge, owing to the substantial gaps in spatial
and temporal scales between different data sources (Fisher et al., 2006; Park et al., 2021).
The practice of conducting manual, in-situ observations for species phenology (SP) boasts a rich history spanning
several centuries (Aono and Kazui, 2008), yielding precise phenological information for the individual plant species (Polgar
and Primack, 2011). In 1963, the Chinese Academy of Sciences inaugurated the Chinese Phenology Observation Network
(CPON), a standardized, nationwide network employing a multitude of professional observers and incorporating extensive
ground-based observations. To date, CPON has amassed over 1.2 million SP records pertaining to more than 900 plant
species across over 150 sites throughout China (Fig. 1), cementing its dominant status as a data center for phenological
research in China. These SP records have been contributed to examining the spatiotemporal patterns of plant phenological
shifts (Dai et al., 2014; Ge et al., 2015), the environmental determinants influencing plant phenology (Dai et al., 2013; Wang
et al., 2020a), as well as the development of phenology models in China (Tao et al., 2018; Wang et al., 2015). Nonetheless,
the spatial coverage of in-situ SP data remains sporadic and restricted on regional and global scales (Donnelly et al., 2022),
with noticeable gaps appearing in longer time scales. The progression of species-level phenology modeling presents an
opportunity to address these limitations (Fu et al., 2020; Hufkens et al., 2018). In the absence of actual observed SP data,
phenology models can be employed to generate large-scale predictions, thereby interpolating the missing SP data in both
space and time (Cleland et al., 2007; Schwartz et al., 2013; Wang et al., 2012). For instance, the Extended Spring Indices
(SI-x) model has been successfully applied to create gridded maps illustrating the first leaf and first bloom events for three
woody plants at a resolution ranging from 1° to 1 km across the contiguous United States (Ault et al., 2015; Izquierdo-
Verdiguier et al., 2018). Similarly, this model-based approach can be adapted to model and map the SP data throughout
China. This would enable the integration and synthesis of CPON's long-term phenology observations at regional and
national scales within the country.
In contrast to manual in-situ observations, satellite remote sensing facilitates the monitoring and mapping of land
surface phenology (LSP) on a more expansive scale. It provides more comprehensive LSP information at the landscape level
(Studer et al., 2007). Over the past four decades, remote sensing technology has witnessed considerable advancements,
significantly improving the spatial and temporal resolution (Misra et al., 2020; Dronova and Taddeo, 2022). At present, a
multitude of LSP products derived from vegetation indices (e.g., NDVI and EVI) procured from multi-source remote sensing
data are accessible, offering regional and global LSP data with varying spatial resolutions ranging from 10 km to 30 m (e.g.,
Ganguly et al., 2010; Li et al., 2019; Wu et al., 2021; Zhang et al., 2020). The credibility of these LSP data remains largely
contingent upon ground phenology (GP) validation based on in-situ observed SP data (Tian et al., 2021; Zhang et al., 2017),
particularly the coordination and aggregation from individual-level phenology (i.e., SP) to landscape-level phenology (i.e.,



GP). Weighted average and quantile methods have been proven effective for aggregating phenology from individual to
community or landscape levels (Donnelly et al., 2022; Fitchett et al., 2015). Prior research has validated weighted average
method on a site scale through field investigations and remote sensing monitoring, to aggregate GP at the community or
landscape levels from in-situ SP data weighted by species abundance (Liang et al., 2011). Some recent studies have
suggested that the quantile method (e.g., 30th percentile) holds greater promise than the commonly used average method on
a larger scale, as evidenced in Europe and the USA (Ye et al., 2022). However, there is no previous study endeavored to
employ these methods for aggregating large-scale GP from SP data in China, which may constrain the availability of ground
validation evidence for LSP products and hinder comprehensive understanding of the spatio-temporal characteristics of
phenological changes over the country.
In this study, we aimed to develop long-term SP and GP maps of China with a 0.1° resolution spanning 1951-2020,
supplying spatially continuous grided phenology products currently absent in the country and crucial for a wider array of
applications. We utilized 24,552 in-situ phenology observations of 24 representative woody plants from 122 sites over the
past six decades from CPON. Three phenophases, namely the first leaf date (FLD), first flower date (FFD), and 100% leaf
coloring date (LCD), were included for each species. We employed five species-level phenology models and grided
meteorological data to simulate and produce SP maps, and utilized species distribution maps as masks of SP maps for each
corresponding plant species. We applied weighted average and quantile methods on SP maps to aggregate and produce GP
maps, which used the distribution probabilities of each species as weights. The accuracy of SP maps was assessed through
cross-validation, while the reliability of GP maps was evaluated by comparing them with existing LSP products. This study
introduces a novel grid phenology dataset for China, which supplements China's existing phenology data sources and
provides an independent phenology data source for LSP product verification. The dataset will facilitate more comprehensive
research on plant phenology and global change by better characterizing the spatiotemporal patterns of plant phenology.

## 2 Methods

### 2.1 Data acquisition and processing

### 2.1.1 Phenology observations

The in-situ phenology observations from 1963 to 2018 were obtained from the CPON. We selected 24 species of woody
plants from 17 families in China (Table 1) that are common and widespread in forest ecosystems in China (Fang et al., 2011)
and well-documented in CPON. These species have been observed over 55 years in 122 sites, with a total of 24,552 records,
covering a range of land cover, ecological, and climatic conditions across China (Fig. 1). Each species had at least 40 years
and 13 sites of phenology data. We studied three phenophases for each species: spring FLD, spring FFD, and autumn LCD.
Outliers were eliminated for each species based on the principle of three sigma limits.

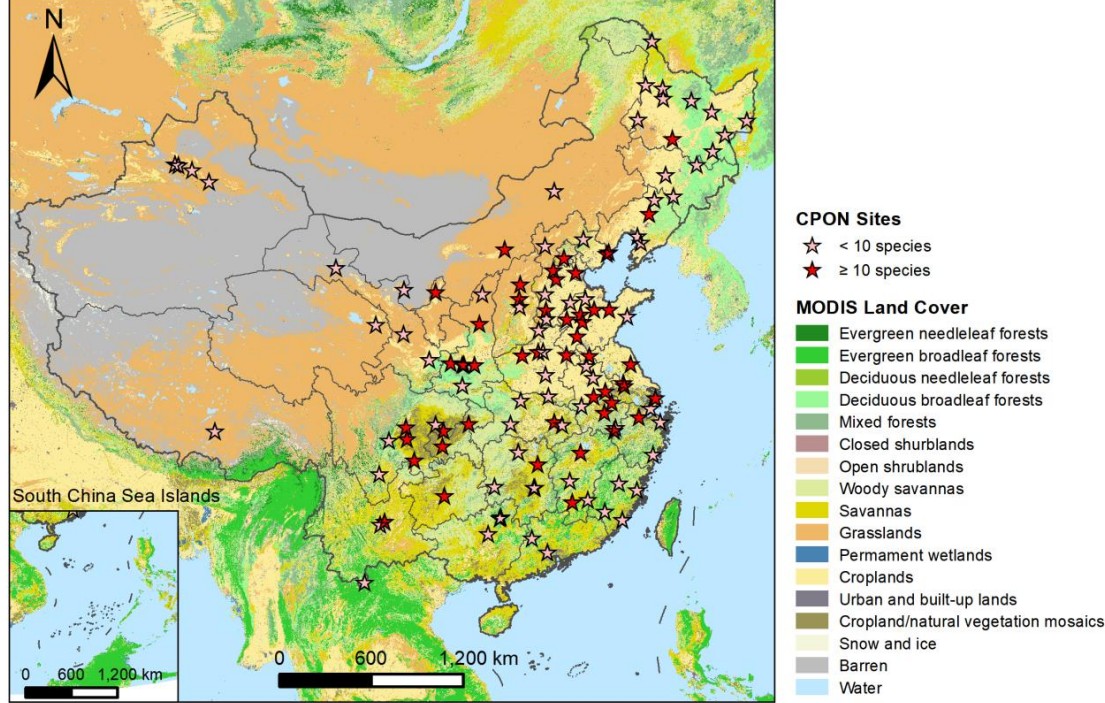


**Figure 1:** Geographic distribution of CPON sites (n = 122) included in the phenology dataset across China. Sites with less than 10 recorded species are marked with pink asterisks, while sites with more than 10 recorded species are marked with red asterisks. Note that the markings on the map of several adjacent sites may overlap each other. The background map shows the IGBP land cover type from the MODIS Land Cover product (Friedl and Sulla-Menashe, 2022).


**Table 1:** List of 24 species of woody plants from 17 families in China. Number of records represents the total number of three phenophases (FLD, FFD and LCD) of all sites and all years for each species.

| No. | Species | Family | Life form | Number of sites | Number of years | Number of records |
|---|---|---|---|---|---|---|
| 1 | *Ginkgo biloba* | Ginkgoaceae | Tree | 45 | 49 | 1110 |
| 2 | *Metasequoia glyptostroboides* | Cupressaceae | Tree | 37 | 47 | 860 |
| 3 | *Magnolia denudata* | Magnoliaceae | Tree | 42 | 47 | 980 |
| 4 | *Salix babylonica* | Salicaceae | Tree | 65 | 42 | 1526 |
| 5 | *Populus × canadensis* | Salicaceae | Tree | 43 | 51 | 954 |
| 6 | *Robinia pseudoacacia* | Fabaceae | Tree | 54 | 45 | 1757 |



| 7 | *Albizia julibrissin* | Fabaceae | Tree | 36 | 47 | 984 |
|---|---|---|---|---|---|---|
| 8 | *Cercis chinensis* | Fabaceae | Shrub | 52 | 49 | 1207 |
| 9 | *Prunus armeniaca* | Rosaceae | Tree | 46 | 45 | 950 |
| 10 | *Ulmus pumila* | Ulmaceae | Tree | 60 | 44 | 1428 |
| 11 | *Morus alba* | Moraceae | Tree | 50 | 50 | 1071 |
| 12 | *Broussonetia papyrifera* | Moraceae | Tree | 41 | 43 | 1103 |
| 13 | *Quercus acutissima* | Fagaceae | Tree | 17 | 40 | 292 |
| 14 | *Pterocarya stenoptera* | Juglandaceae | Tree | 29 | 46 | 936 |
| 15 | *Juglans regia* | Juglandaceae | Tree | 50 | 47 | 816 |
| 16 | *Betula platyphylla* | Betulaceae | Tree | 13 | 43 | 369 |
| 17 | *Acer pictum* subsp. *mono* | Sapindaceae | Tree | 18 | 46 | 492 |
| 18 | *Ailanthus altissima* | Simaroubaceae | Tree | 34 | 47 | 873 |
| 19 | *Melia azedarach* | Meliaceae | Tree | 61 | 46 | 1410 |
| 20 | *Firmiana simplex* | Malvaceae | Tree | 57 | 48 | 1403 |
| 21 | *Hibiscus syriacus* | Malvaceae | Shrub | 58 | 47 | 1096 |
| 22 | *Fraxinus chinensis* | Oleaceae | Tree | 23 | 40 | 505 |
| 23 | *Syringa oblata* | Oleaceae | Shrub | 50 | 51 | 1163 |
| 24 | *Paulownia fortunei* | Paulowniaceae | Tree | 49 | 48 | 1267 |
| Total | | - | - | 122 | 55 | 24552 |


### 2.1.2 Climate data

The daily mean temperature (T) from 1950-2020 were obtained from two sources: (1) Site T was extracted from climate
observations in the China Meteorological Data Service Center (CMDSC, https://data.cma.cn/) and used to parameterize the
phenology models. (2) Grid T was extracted from ERA5-Land climate reanalysis data (Muñoz Sabater, 2019; Muñoz-
Sabater et al., 2021) from the Copernicus Climate Change Service (C3S, https://cds.climate.copernicus.eu/) and used for
phenology simulation and upscaling at a spatial resolution of 0.1° (about 10 km). Hourly grid T was averaged across four
phases (4:00, 10:00, 16:00, 22:00) to derive the daily grid T.
The current bioclimatic (BIOCLIM+) variables were obtained from Climatologies at High Resolution for the Earth
Land Surface Areas (CHELSA, https://chelsa-climate.org/) to determine the species distribution (Brun et al., 2022a, b). The





BIOCLIM+ variables indicate the average ecological and climatic conditions during 1981-2010, with a high resolution of
0.0083°. We extracted the traditional 19 bioclimatic layers (Bio1-Bio19) and the complementary 50 bioclimatic layers in
China. We calculated the correlation between every two layers to reduce the impact of autocorrelation among these
bioclimatic layers, and then excluded the layers with a correlation coefficient greater than 0.8 with the previous layers. As a
result, 12 bioclimatic layers were retained as the environmental data inputs for the species distribution models (Table S1).
These layers were resampled to 0.1° to match the resolution of the grid T data.

### 2.1.3 Forest and species distribution data

The forest distribution map of China was derived from the dataset of "Annual Dynamics of Global Land Cover and its
Long-term Changes from 1982 to 2015" (Liu et al., 2020). Each year's land cover (LC) layers were reclassified as forest and
non-forest, and then the number of years of forest cover was obtained by adding all layers. Pixels with at least one year of
forest cover were identified as forest distribution areas. The forest distribution map was resampled from 0.05° to 0.1° by the
majority method to match the resolution of the grid T data.
The county-level species distribution maps were obtained from the updated Database of China's Woody Plants (Fang et
al., 2011). The distribution maps in this database were compiled from all national, provincial, and regional floras and
inventory reports in China published before 2009, which are considered authoritative (Cai et al., 2021). We then obtained the
species occurrence records from the Global Biodiversity Information Facility (GBIF; https://www.gbif.org/), and used them
as the occurrence data inputs for the species distribution models (GBIF, 2022). The occurrence records were filtered by
including the coordinate locations with uncertainty less than 2000 meters, and cleaned by removing duplicate records.

### 2.2 Generating species phenology maps using a model-based upscaling method

The generation of species phenology maps involves two major processes: (1) Generating species potential phenology
maps, and (2) Generating species distribution maps. The final SP maps were obtained by spatially intersecting these two
maps. The workflow for the processes is shown in Fig. 2.

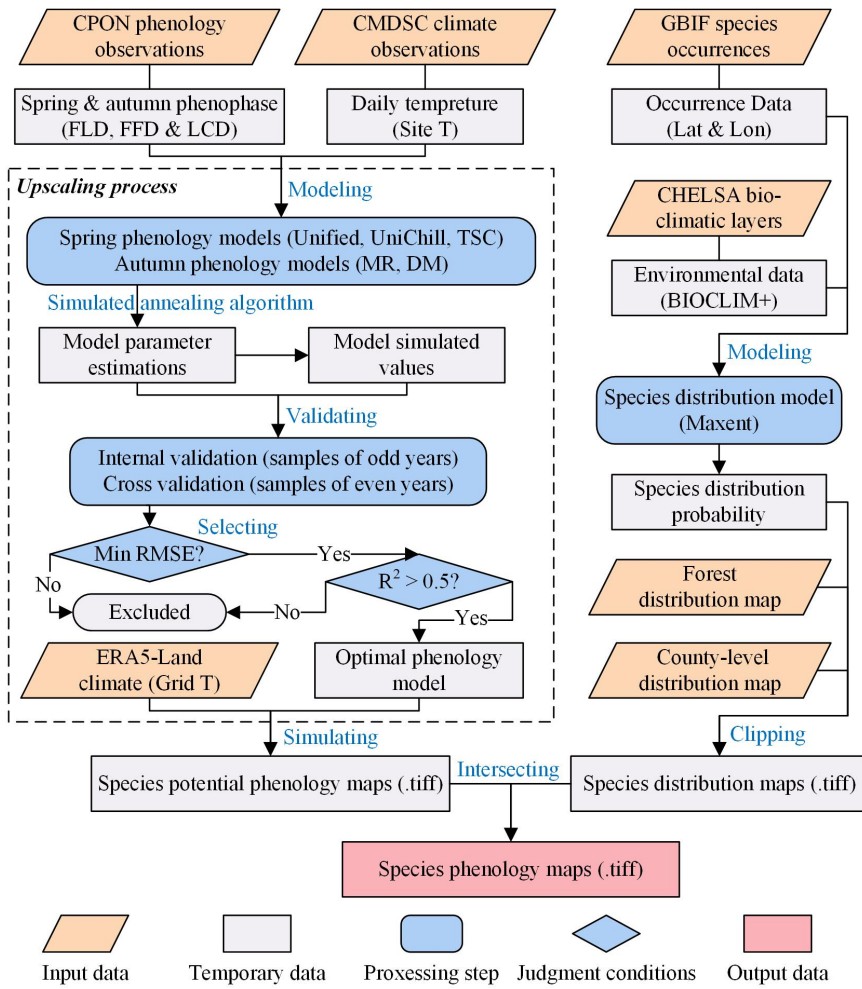


**Figure 2:** The workflow of generating SP maps using a model-based upscaling method, which involves two major processes: (1) Generating species potential phenology maps, and (2) Generating species distribution maps. The words in blue color represent the key processes of data generation. ".tiff" indicates the GeoTIFF format of the grid phenology or distribution maps.

**2.2.1 Species potential phenology maps**

In the first process, we used a model-based upscaling method to convert in-situ phenology observations into grid phenology maps. Phenology models were built using the phenophases (i.e., FLD, FFD, LCD) from CPON phenology observations and the corresponding site T from CMDSC climate observations. For each species, we built three spring phenology models: the Unichill, Unified (Chuine, 2000) and temporal-spatial coupling (TSC) models (Ge et al., 2014), and two autumn phenology models: the multiple regression (MR) (Estrella and Menzel, 2006) and temperature-photoperiod (TP)



models (Delpierre et al., 2009). The details of the model formulae are described in Appendix S1. For each model, samples
from odd years were used for phenology modeling, and samples from even years were reserved for cross validation on the
model. All model parameters were estimated using the simulated annealing algorithm (Chuine et al., 1998).
For model validation, the models' root mean square error (RMSE) and goodness of fit ($R^2$) were calculated between the
model simulated values and original values. Internal validation was conducted on samples from odd years to evaluate the
fitting effect of the model, and cross validation was conducted on samples from even years to evaluate the simulation and
extrapolation effect of the model. The optimal phenology model for each species was selected based on the smallest RMSE
in cross validation and $R^2$ greater than 0.5 (0.3 for LCD) in both validations. If no model met these conditions, the species
was excluded when generating SP maps or GP maps.
For simulating SP maps, daily grid T data from ERA5-Land climate reanalysis were input into the optimal phenology
model and simulated pixel by pixel. This way, the phenology observations from individual sites were interpolated and
upscaled into a grid phenology map based on the phenology models (Chuine et al., 2000). However, as long as there was
grid T data, simulated species phenology could be obtained, even if there was no species distribution. Therefore, we named it
as species potential phenology map to avoid taking simulated values as true values in areas without species distribution.
**2.1.2 Species distribution maps**
In the second process, we simulated the species distribution maps using both species distribution models and county-
level species distribution data. Species distribution models were built for each species using Maximum Entropy Species
Distribution Modelling (Maxent; Phillips et al., 2006) version.3.4.4. Maxent estimates the range of a species by finding the
species distribution of maximum entropy (i.e., closest to the uniform), which is widely adopted in species distribution
modeling (Phillips et al., 2006). It expresses a probability distribution where each grid cell has a predicted probability of
presence for the species. To build the Maxent model, species location records from the GBIF database were used as
occurrence data input, and the 12 bioclimatic layers from BIOCLIM+ were used as the environmental data input. In the
model parameter settings, linear and quadratic feature types were used and 5-fold cross validation was used as the replicated
run type.
For model validation, the receiver operating characteristic (ROC) curve analysis method was used to test the accuracy
of the Maxent prediction model. The area under the ROC curve, known as the AUC value, is usually used as an indicator of
the prediction accuracy of the model (Fielding and Bell, 1997). The closer the AUC value is to 1.0, the more accurate the
prediction result of the model is. The average test AUC for different species was 0.845, with a standard deviation of 0.043.
**2.3 Generating ground phenology maps using weighted average and weighted quantile methods**
We used four methods to aggregate from individual-level SP maps to landscape-level GP maps: (1) weighted average
(mean); (2) weighted median (pct50); (3) weighted 20th percentile (pct20) for spring phenology or weighted 80th percentile
(pct80) for autumn phenology; (4) weighted 10th percentile (pct10) for spring phenology or weighted 90th percentile (pct90)
for autumn phenology. The weight of each species was determined by the species distribution probability, as it is assumed
that the species abundance is positively related to the species distribution probability. The aggregation methods of GP in this
study (e.g., pct50, pct20\80 and pct10\90) are comparable and similar to the extraction methods of LSP from remote sensing
data (e.g., midpoint, dynamic threshold and maximum curvature). The workflow is shown in Fig. 3.

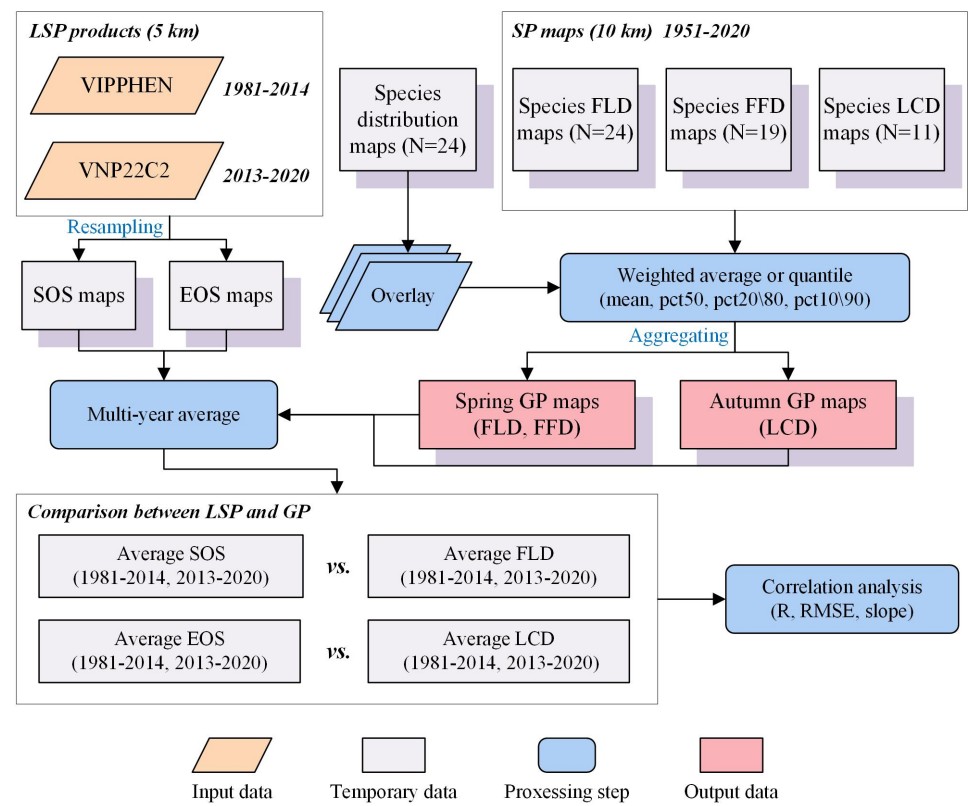


**Figure 3:** The workflow of generating GP maps from SP maps, and comparing GP maps with two LSP products. The words
in blue color represent the key processes of data generation.

For $n$ species, the phenophases ($Y$) were first sorted from small to large. The SP of each species is $y_i$ ($i = 1, 2, \ldots, n$),
and the distribution probability of each species is $p_i$ ($i = 1, 2, \ldots, n$). Then, the aggregated GP ($Y_{mean}$ and $Y_{pct}$ ($x\%$)) was
calculated according to the following formulas:
$\omega_i = \frac{p_i}{\sum_{i=1}^{n} p_i}$   (1)
$W_j = \sum_{i=1}^{j} \omega_i, j = 1, 2, \ldots, n$   (2)
$Y_{mean} = \sum_{i=1}^{n} \omega_i \times y_i$   (3)



$$Y_{pct} = \begin{cases} y_1, if\ W_1 > x \\ (y_j - y_{j-1}) \times \frac{x - W_{j-1}}{\omega_j}, if\ W_j > x, W_{j-1} < x \\ y_n, if\ W_{n-1} < x \end{cases}$$
(4)

Where $\omega_i$ is the weight of each species, $W_j$ is the cumulative weight from the first to the $j$ species, $x\%$ is the percentile tag
which takes values from 10%, 20%, 50%, 80% and 90%. These formulas were used to calculate the aggregated GP maps by
combining the species phenology maps with the species distribution maps and weighting them by the species distribution
probability.
Finally, to assess data quality, the aggregated GP maps in this study were compared with two LSP products extracted
from remote sensing in previous studies to assess data quality: (1) VIPPHEN_NDVI product (1981-2014), which used
midpoint method to extract the start of season (SOS) and the end of season (EOS) from the AVHRR data (Didan and
Barreto, 2016); (2) VNP22C2 product (2013-2020), which used maximum curvature method to extract SOS and EOS from
the MODIS data (Zhang et al., 2020b). Both LSP products were resampled from 5 km to 0.1° by the average method to
match the spatial resolution of GP maps. The LSP and GP maps were averaged in two segments (1981-2014 and 2013-
2020), and the correlation analysis was conducted between FLD and SOS in spring and between LCD and EOS in autumn.
Pearson correlation coefficient (R), RMSE, and linear regression slope were used to evaluate the consistency between GP
and LSP.

## 3 Results and discussion

The dataset includes two types of phenology maps over China: (1) Yearly SP maps generated by the model-based
upscaling method for 24 woody plants; (2) Yearly GP maps generated by four aggregation methods, along with the
corresponding quality assurance (QA) maps. The phenology maps provide spring FLD, FFD, and autumn LCD of woody
plants and forests over China from 1951 to 2020, with a spatial resolution of 0.1° and a temporal resolution of 1 day. Each
map is stored in a 16-bit signed integer file in GeoTIFF format, which contains a two-dimension raster (641 row × 361
column). The unit of phenology data is the Julian Day of year (DOY), which represents the actual number of days from
January 1st to the date of phenology occurrence. The valid values range from DOY 1 to 366, and the null values equal to -1.

### 3.1 Simulation and validation of species phenology maps

The SP maps of FLD (24 species), FFD (19 species), and LCD (12 species) were simulated using the optimal
phenology models, and then masked by the species distribution maps. Here, we present the results of simulated SP maps of
four typical woody species (Fig. 4), including ginkgo (*Ginkgo biloba*), willow (*Salix babylonica*), elm (*Ulmus pumila*), and
lilac (*Syringa oblata*). These maps showed that the phenophases of different species have a consistent spatial pattern of
variation along latitude. Specifically, spring FLD and FFD of these species were significantly later with increasing latitude,



while autumn LCD was significantly earlier with increasing latitude. Despite similar spatial patterns, the phenophases of
different species show distinct temporal differences at the same latitude; for example, at lower latitudes, elm has
significantly earlier spring FFD and later autumn LCD than other species. Phenophases of some species were not simulated
because the $R^2$ of their optimal models was too small, e.g., $R^2 < 0.5$ for spring FFD, and $R^2 < 0.3$ for autumn LCD.


**Figure 4:** Species phenology (SP) maps of four typical woody species averaged from 1951 to 2020. Columns 1-2 show the
spring phenophases (FLD and FFD), and Column 3 shows the autumn phenophase (LCD). Each row represents a species
from ginkgo (*Ginkgo biloba*), willow (*Salix babylonica*), elm (*Ulmus pumila*), and lilac (*Syringa oblata*). The unit of SP is
the Julian Day of year (DOY) from January 1st.



**Table 2:** The optimal phenology models and cross-validation results of 24 species. RMSE represents the root mean square error between the model simulated values and original values. $R^2$ represents goodness of fit of the optimal phenology model.

| No. | Species | FLD | | | FFD | | | LCD | | |
|---|---|---|---|---|---|---|---|---|---|---|
| | | Optimal model | RMSE | $R^2$ | Optimal model | RMSE | $R^2$ | Optimal model | RMSE | $R^2$ |
| 1 | *Ginkgo biloba* | TSC | 7.30 | 0.669 | TSC | 7.53 | 0.553 | DM | 12.54 | 0.401 |
| 2 | *Metasequoia glyptostroboides* | TSC | 6.10 | 0.687 | Unified | 9.59 | 0.126 | DM | 9.99 | 0.295 |
| 3 | *Magnolia denudata* | UniChill | 6.47 | 0.781 | TSC | 7.33 | 0.576 | DM | 9.31 | 0.284 |
| 4 | *Salix babylonica* | TSC | 8.97 | 0.854 | TSC | 9.40 | 0.787 | MR | 18.23 | 0.380 |
| 5 | *Populus × canadensis* | UniChill | 5.94 | 0.808 | UniChill | 6.14 | 0.728 | MR | 9.45 | 0.139 |
| 6 | *Robinia pseudoacacia* | TSC | 5.47 | 0.863 | TSC | 6.18 | 0.785 | DM | 11.74 | 0.297 |
| 7 | *Albizia julibrissin* | UniChill | 7.48 | 0.500 | Unified | 8.23 | 0.376 | MR | 9.18 | 0.567 |
| 8 | *Cercis chinensis* | TSC | 7.90 | 0.723 | UniChill | 7.39 | 0.751 | DM | 9.09 | 0.175 |
| 9 | *Prunus armeniaca* | TSC | 6.05 | 0.865 | UniChill | 4.78 | 0.929 | MR | 14.52 | 0.191 |
| 10 | *Ulmus pumila* | UniChill | 5.09 | 0.901 | UniChill | 8.38 | 0.862 | DM | 11.16 | 0.654 |
| 11 | *Morus alba* | TSC | 6.70 | 0.905 | UniChill | 7.99 | 0.860 | DM | 9.04 | 0.175 |
| 12 | *Broussonetia papyrifera* | UniChill | 7.60 | 0.804 | TSC | 6.18 | 0.821 | DM | 9.97 | 0.615 |
| 13 | *Quercus acutissima* | UniChill | 6.73 | 0.931 | UniChill | 5.12 | 0.950 | MR | 14.35 | 0.765 |
| 14 | *Pterocarya stenoptera* | UniChill | 7.52 | 0.804 | UniChill | 7.89 | 0.710 | MR | 11.57 | 0.415 |
| 15 | *Juglans regia* | TSC | 6.04 | 0.739 | UniChill | 8.54 | 0.595 | DM | 8.41 | 0.141 |
| 16 | *Betula platyphylla* | UniChill | 3.80 | 0.915 | UniChill | 3.70 | 0.906 | DM | 8.27 | 0.655 |
| 17 | *Acer pictum* subsp. *mono* | TSC | 2.29 | 0.894 | TSC | 3.78 | 0.814 | DM | 4.71 | 0.670 |
| 18 | *Ailanthus altissima* | UniChill | 5.22 | 0.867 | UniChill | 8.34 | 0.664 | DM | 10.39 | 0.066 |
| 19 | *Melia azedarach* | TSC | 6.81 | 0.828 | TSC | 6.70 | 0.851 | MR | 10.19 | 0.135 |
| 20 | *Firmiana simplex* | UniChill | 6.02 | 0.694 | Unified | 8.10 | 0.314 | DM | 12.30 | 0.190 |
| 21 | *Hibiscus syriacus* | TSC | 9.66 | 0.666 | Unified | 13.38 | 0.331 | DM | 12.76 | 0.464 |
| 22 | *Fraxinus chinensis* | TSC | 6.25 | 0.852 | Unified | 12.35 | 0.319 | MR | 9.76 | 0.533 |
| 23 | *Syringa oblata* | UniChill | 7.01 | 0.864 | UniChill | 5.11 | 0.920 | MR | 12.36 | 0.475 |




| 24 | *Paulownia fortunei* | UniChill | 4.63 | 0.762 | UniChill | 7.02 | 0.693 | MR | 10.01 | 0.250 |

The simulation effects of species phenology maps were evaluated by cross-validation on the optimal phenology models (Table 2). The results showed that the simulation effects of spring phenology were significantly better than that of autumn phenology (Fig. 5). Specifically, the RMSE of the optimal model of FLD (6.38 days) and FFD (7.46 days) in spring were significantly smaller than that of LCD (10.80 days) in autumn. And the $R^2$ of the optimal model of FLD (0.799) and FFD (0.676) in spring were significantly greater than that of LCD (0.372) in autumn. However, there was no significant difference between FLD and FFD simulation effects in spring. UniChill and TSC models, as the optimal model, had significantly better FFD simulation effects than Unified models for the different phenology models in spring. MR and TP models had similar LCD simulation effects for the different phenology models in autumn.

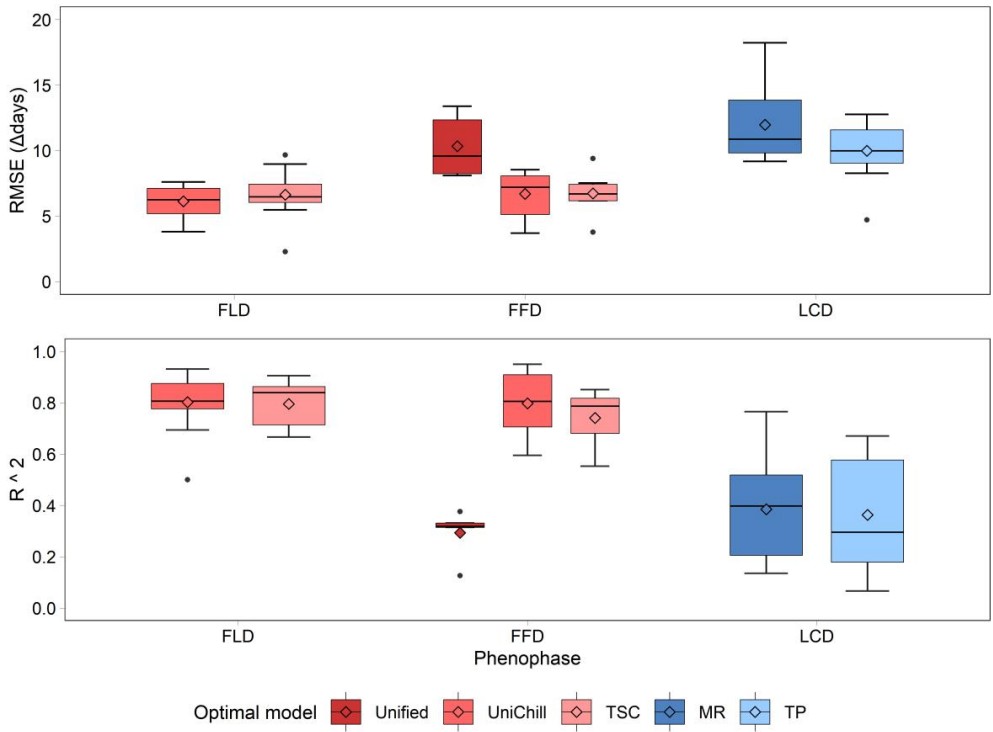

**Figure 5:** The RMSE (a) and $R^2$ (b) of cross-validation on the optimal phenology models for 24 woody species. Each model is represented by a different color, with warm colors for three spring phenology models (Unified, UniChill, TSC), and cool colors for two autumn phenology models (MR, TP). The model with the smallest RMSE was selected as the optimal model for each species. The horizontal line represents the median value, the diamond mark represents the mean value, and the dot mark represents the outlier in the boxplot.



## 3.2 Aggregation of ground phenology maps

The results of GP maps generated by four different aggregation methods (mean, pct50, pct20\80, pct10\90) showed similar spatial patterns (Fig. 6), i.e., the consistent variation along latitude or altitude. With the increase of latitude or altitude, the spring GP (FLD and FFD) became later, and the autumn GP (LCD) became earlier. For different aggregation methods, the GP maps aggregated from the mean and pct50 methods were highly consistent, with R being 0.992; while the GP maps aggregated from the pct20\80 and pct10\90 methods were slightly different from the former two, with R being 0.968 and 0.949, and showed larger spatial variation than the former two. The high consistency between the mean and pct50 maps indicated that both the weighted mean method and weighted quantile method were robust for the aggregation of GP.



**Figure 6:** Ground phenology (GP) maps of four aggregation methods averaged from 1951 to 2020. Columns 1-2 show the spring phenophases (FLD and FFD), and Column 3 shows the autumn phenophase (LCD). Each row represents an aggregation method from weighted average (mean), weighted median (pct50), weighted 20% or 80% percentile (pct20\80), and weighted 10% or 90% percentile (pct10\90). The unit of GP is the Julian Day of year (DOY) from January 1st.

We also provided two QA maps to evaluate the reliability of the aggregation results of GP maps (Fig. S1). The first is the total distribution probability of all species, and the second is the total number of species with distribution probabilities greater than 0.1. In the QA maps, higher values mean larger total number or probability of species for the aggregation, indicating that GP maps have higher reliability in these areas. The regions with the most reliable GP aggregation results were distributed around 30° N in China. The total number of species is about 15 for FLD and FFD, and is about 6 for LCD in these regions. It should be noted that in the QA map, in areas where the total number of species is less than 5 or the total probability of species is less than 1, the aggregation results of GP may not be reliable.

## 3.3 Data quality and usability

GP and LSP were compared between FLD and SOS in spring and between LCD and EOS in autumn during two segments (1981-2014 and 2013-2020). The results showed that GP and two LSP products had similar spatial patterns in central and northern China but relatively different patterns in southern China (Fig. 7), particularly for LCD and EOS in autumn (Fig. 7e-h). This is likely due to the prevalence of deciduous forests (DF) in central and northern China (Fig. 1). In contrast, evergreen forests (EF) and mixed forests (MF) are found in southern China. GP in this study was generated by aggregating the SP of 24 deciduous woody plants, which made up a large proportion of DF but a small proportion of EF or MF. Additionally, LSP extracted from remote sensing data tends to have a larger error in EF and MF due to the lack of obvious seasonal change and frequent cloud cover in these regions (Liu et al., 2016b). As a result, the consistency between GP and LSP was relatively poor in EF and MF areas (Fig. S2), with the maximum R being 0.44 in spring and 0.54 in autumn, and the minimum RMSE being 28.5 days in spring and 38.5 days in autumn (Table S2). In contrast, the consistency between GP and LSP was much better in DF area, with the maximum R being 0.95 in spring and 0.88 in autumn, and the minimum RMSE being 8.8 days in spring and 15.1 days in autumn, respectively.

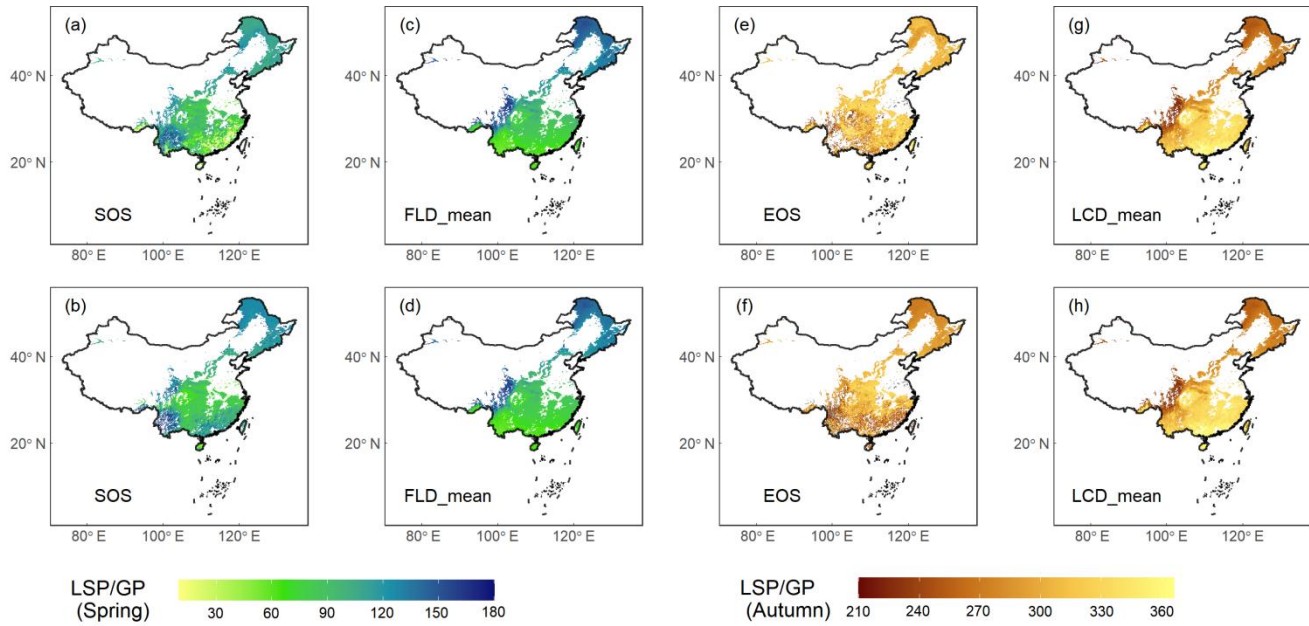

286

**Figure 7:** Comparison of GP maps in this study and two LSP products (VIPPHEN and VNP22C2) extracted from remote sensing in previous studies, which was made between FLD and SOS in spring and LCD and EOS in autumn. Row 1 shows the comparison between VIPPHEN product and GP map averaged in 1981-2014, and Row 2 shows the comparison between VNP22C2 product and GP map averaged in 2013-2020. (a-b) SOS from two LSP products; (c-d) FLD aggregated by mean method; (e-f) EOS from two LSP products; (g-h) LCD aggregated by mean method. The unit of GP or LSP is the Julian Day of year (DOY) from January 1st.

To further assess the quality of the data, we examined the consistency between GP and LSP specifically in DF areas. The results showed that GP and LSP had good consistency in DF areas for both VIPPHEN and VNP22C2 products, i.e., high correlation (R), small difference (RMSE), and good linear relationship (Fig. 8). Compared with the LSP of VIPPHEN product, the LSP of VNP22C2 product has better consistency with the GP of this study. In addition, for both products, the consistency between GP and LSP in spring (Fig. 8e, g) was significantly better than that in autumn (Fig. 8f, h). When comparing different aggregation methods (mean, pct50, pct20/80, pct10/90), there was no significant difference in R between GP and LSP (Fig. 8a, b). All methods produced similar R values, ranging from 0.76-0.78 in spring and 0.49-0.53 in autumn for the VIPPHEN product, and from 0.90-0.91 in spring and 0.79-0.84 in autumn for the VNP22C2 product. However, different methods produced significantly different RMSE values between GP and LSP (Fig. 8c, d), largely due to the differences in the average values of GP under different methods. The best aggregation methods, with the minimum RMSE, were pct10 (20.8 days) in spring and pct90 (32.9 days) in autumn for the VIPPHEN product, and pct20 (8.8 days) in spring and pct90 (15.1 days) in autumn for the VNP22C2 product.

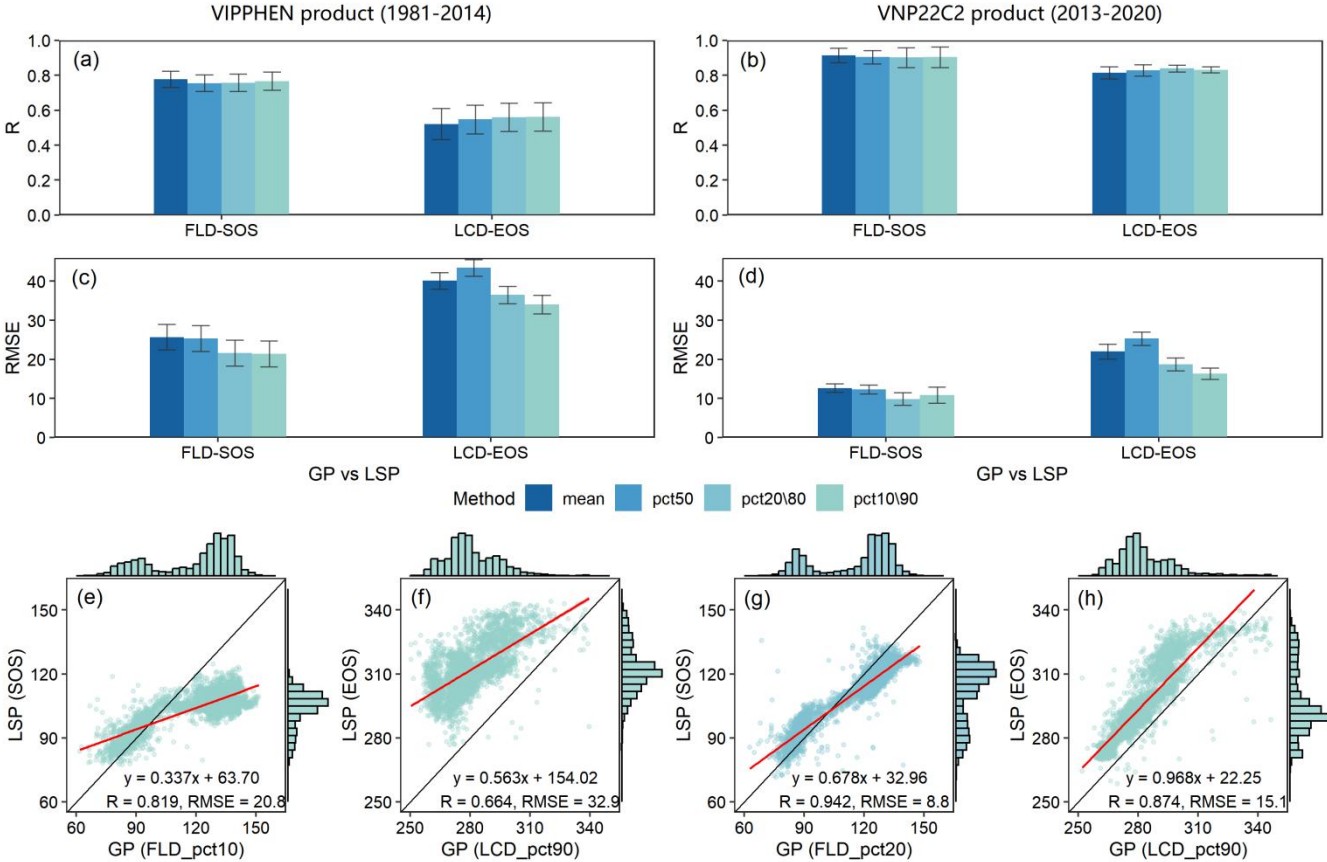

**Figure 8:** Comparison results of GP maps and two LSP products (VIPPHEN and VNP22C2) in DF areas, which was made between FLD and SOS in spring and LCD and EOS in autumn within the time range 1981-2014 and 2013-2020. (a-b) R between LSP and GP under four aggregating methods; (c-d) RMSE between LSP and GP under four aggregating methods; (e-h) Linear relationship between between LSP and GP under the best aggregating method. Each aggregating method is represented by a different color. The best aggregating method was determined by minimizing the RMSE between GP and LSP in DF areas. The error bar in the bar plot represents the multi-year standard deviation. The red line in the scatter plot represents the linear regression line between GP and LSP, and all regression results were extremely significant (p<0.001).

It is worth noting that the aggregation method with the smallest difference between GP and LSP in this study was the 10th or 20th percentile in spring and the 90th percentile in autumn. It means that the spring green-up event detected by remote sensing is more consistent with the FLD of earlier-developing plant species (the first 10%-20%) on the ground, while the autumn dormancy event from remote sensing is more consistent with the LCD of later-senescent plant species (the last 10%) on the ground. These results reveal a potential connection between GPs and LSPs despite their different physical implications in diagnosing phenology.



In general, this dataset provides high reliability SP and GP simulations of forests over China for the past 70 years. It is
an independent phenology data source generated by the modeling and aggregation based on ground observations. There are
several considerations in data application:
(1) For SP maps, the accuracy of data was determined by RMSE and $R^2$ of cross-validation on the optimal phenology
model for each species (Table 2). Additionally, the reliability of SP in space was affected by the number of sites available for
modeling on each species (Table 1). For instance, the accuracy of *Betula platyphylla*'s FLD was very high overall (RMSE =
3.80 and $R^2$ = 0.915), but the local accuracy might be relatively poor in areas with sparse sites due to very few sites of *Betula*
*platyphylla* in space (n = 13). In this study, the SP maps of 24 species in China were found to be largely consistent with the
in-situ observations, with an average error of 6.4, 7.5 and 10.8 days for FLD, FFD and LCD, respectively. These errors were
the same or smaller than those of phenology modelling in previous studies. For example, the simulation error of spring FLD
and FFD was 7-9 days in central Europe (Basler, 2016) and was 12.3-12.7 days in the United States (Izquierdo-Verdiguier et
al., 2018), while the simulation error of autumn LCD was 10.3-13.0 days in France (Delpierre et al., 2009) and 5.9-22.8 days
in the United States (Jeong and Medvigy, 2014). Therefore, compared with other studies on the regional scale, the SP maps
of China in this study were found to have relatively high accuracy.
(2) For GP maps, the reliability of data can be determined by QA maps which provide the total number or probability of
species. Additionally, the reliability can also be evaluated by comparing GP data with other LSP products, with high
consistency indicating good reliability. Since GP data actually provide phenology estimates of the DF components in the
forests, it has better reliability in the DF areas but less reliability in EF or MF areas. In this study, GP maps of forests in
China were found have good consistency with the existing LSP products, particularly in DF areas, where the correlation
coefficients of FLD and LCD were 0.91 and 0.84, respectively. The differences between GP and LSP in FLD and LCD were
also found to be relatively small in DF areas, being 8.8 days and 15.1 days, respectively. Previous studies have shown poor
consistency between single species and LSP, with correlation coefficients ranging from 0.50 to 0.51 in the United States
(Peng et al., 2017) and Germany (Kowalski et al., 2020), and differences ranging from 12 to 14.5 days in the United States
(Peng et al., 2017) and Canada (Delbart et al., 2015). In contrast, research comparing average or quantile values of multiple
species has shown better results similar to this study. For example, the correlation coefficients between the average (or
weighted average) GP and LSP were found to be 0.61 to 0.71 in Europe (Rodriguez-Galiano et al., 2015; Tian et al., 2021).
The correlation coefficients between the 30th percentile GP and LSP were found to be 0.54 to 0.57 in China (Wu et al.,
2016). The differences between the GP and LSP in previous studies were 10.3-12.4 days in China (Wu et al., 2016), 13.9
days in Europe, and 12.3 days in the United States (Ye et al., 2022), which was larger than the results of FLD but smaller
than that of LCD in this study. Although the landscape-level GP data aggregated from species-level SP data in this study
showed good reliability, limitations in available species and different aggregation methods suggest that future comparisons
between GP and LSP in other regions still need to be improved.
(3) For phenology maps in different seasons, the reliability of phenology data in spring was found to be significantly
higher than that in autumn. The underlying reason is that the mechanism of autumn phenology is more complex compared to





that of spring phenology (Menzel, 2002). Moreover, the influencing factors of autumn phenology are not yet fully
understood, which poses an additional challenge (Gill et al., 2015; Wu et al., 2018). In addition to temperature, other
environmental factors such as precipitation (An et al., 2020), photoperiod (Lang et al., 2019), solar radiation (Wu et al.,
2021b), spring phenology (Liu et al., 2016a), and growing-season productivity (Zani et al., 2020) may also drive autumn
phenology. Thus, modeling autumn phenology is more challenging compared to spring phenology (Melaas et al., 2016),
resulting in poorer model performance and inferior data quality of SP or GP maps in autumn.

**4 Data availability**

The annual SP and GP maps over China can be accessed at https://doi.org/10.57760/sciencedb.07995 (Zhu et al., 2023).
This dataset is licensed under a CC-BY 4.0 license. The spatial reference system of the dataset is EPSG:4326(WGS84).

**5 Conclusions**

In this study, mainly based on CPON historical phenology observations, we developed a new long-term gridded
phenology dataset: SP maps of 24 woody plants and GP maps of forests over China from 1951–2020, with a spatial
resolution of 0.1° and a temporal resolution of 1 day. For the generation of SP maps, we adopted a model-based upscaling
method to realize the scale expansion of SP date from in-situ to regional scales in China. For the generation of GP maps, we
adopted weighted average and weighted quantile methods to realize the aggregation from species to community or landscape
levels in China. Dataset quality assessment shows that the average error of SP maps is 6.9 days in spring and 10.8 days in
autumn, and the minimum difference between GP maps and existing LSP products is 8.8 days in spring and 15.1 days in
autumn. Compared to the previous studies (Basler, 2016; Delpierre et al., 2009; Izquierdo-Verdiguier et al., 2018; Jeong and
Medvigy, 2014; Tian et al., 2021; Wu et al., 2016; Ye et al., 2022), the SP maps in this study have the same or smaller
simulation error, and the GP maps in this study have good agreement with other LSP products, so the data has high accuracy
and reliability. This dataset is the first phenology map of China. It can be used to investigate the spatial pattern of plant
phenology more clearly along the geographic gradient (e.g., longitude, latitude, and altitude), and to reveal the temporal
trends (e.g., interannual, decadal, and secular) of plant phenology across China. The dataset can also provide important data
support for global change impact assessment, terrestrial ecosystem simulation, and natural resource management.

**Author contribution**

QG and JD designed the study and planned the modeling. HW developed the model code. WL and YH performed the
simulations. MZ processed the modeling data, performed the computations and drafted the manuscript. JD and JA critically
revised the manuscript. All authors discussed and contributed to the modeling and manuscript.



**Competing interests**

The authors declare that they have no conflict of interest.

**Acknowledgements**

This study was jointly supported by National Key Research and Development Program of China (2018YFA0606102), National Natural Science Foundation of China (42271062), and Strategic Priority Research Program (A) of Chinese Academy of Sciences (XDA19020303; XDA26010202). Phenology data was provided by CPON. Temperature data was provided by Copernicus Climate Change Service (C3S).

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
