# Peer review of "Mapping 24 woody plant species phenology and ground forests phenology over China from 1951-2020"

_Earth System Science Data, 2023_

## Author Response (AR1)

Dear Editor,

We are trying to respond to all the reviewers' comments in details as below. We have revised the manuscript and addressed almost all of the points raised by the referees, especially on what referee have pointed out about the justification of using species distribution probability as weight in aggregation method. We have added more information on the outlier removal method and the aggregation method, added a supplement table, added two references as instructed, clarified the expressions, and modified some abbreviations and formats in this revision.

Many thanks for all the work all the referees and the editors have done on this manuscript. We hope this revision can solve most of the problems you referred to in your comments.

Please find the specifics below.

**Response to reviewers & editors' comments**

**Reviewer: 1**

**Comments from the Referee**
In this study the authors properly used phenological models to map 24 woody plant species phenology based on 24552 in-situ phenological observation records in China. What impressed me was that they used so many ground observation records conducted by China Phenological Observation Network (CPON), and applied weighted average and quantile methods to aggregate and to map ground forests phenology over China for the past 70 years. By this manuscript, the authors provided, for the first time, phenological atlas of China based on ground observation data over a long period of time. The aggregation method adopted in this paper was novel and strong, and the simulation results of phenological data were believable compared to data derived from land surface phenological products. The dataset presented in this manuscript has a very broad application prospects and is worthy of fast publication. However, I have some minor points need to be addressed by the authors as the following.

**(1) In section 2.1.3:** "Forest and species distribution data", you did not mention the data sources on which deciduous, evergreen, and mixed forests are classified in your work. I suggest you tell the data source in details and tell its application rationality in this work.
**Response:**
Thank you very much for your mention of the data sources for deciduous, evergreen, and mixed forest distribution. In this study we used the most commonly used International Geosphere-Biosphere Program (IGBP) classification from MODIS Land Cover Type (MCD12C1) to identify different types of forests. We merged evergreen needleleaf forest (class 1) and evergreen broadleaf forest (class 2) into evergreen forest (EF), and deciduous needleleaf forest (class 3) and deciduous broadleaf forest (class 4) into

deciduous forest (DF). Mixed forest (class 5) was also included in the forest type.

**Changes in manuscript:**

We have added an introduction on Lines 125-130, explaining the data source and application of the forest type.

**(2) Line 128-129:** you did not tell how many species occurrence records were used here, so I recommend you add such information.

**Response:**

We agree with the referee's suggestions at this.

**Changes in manuscript:**

We have added the total numbers of occurrence records (4371) for the 24 species used in this study on Line 134, and have added Table S2 in the supplementary material with the to specific number of occurrence records used for each species. This will help to better present simulated data and results of species distribution in this work.

**(3) Line 199-200:** the phrase "to assess data quality" appears twice. Please modify.

**Response:**

Thank you very much for your reminder at this.

**Changes in manuscript:**

We have removed the second repetition of the phrase "to assess data quality" (Line 210) and kept the first one (Line 209).

**(4) Line 241-243:** the names of different models and the meaning of "for the different phenology models" were not expressed properly. Please delete the latter.

**Response:**

Yes, we agree with this suggestion about expression.

**Changes in manuscript:**

We have deleted the original statement and revised it to the following sentence (Line 250-252): "Among the optimal spring phenology models, the FFD simulation effects of UniChill and TSC models were significantly better than Unified model. But in autumn, the LCD simulations effects are similar for MR and TP models."

**(5) Figure 7** looks a little confusing, as the names of the legend system in figure 7(c) and figure 7(d) were the same and the appearance of both figures were very similar, and so it was with figure 7(g) and figure 7(h). Please clarify their differences by marking the year for the data in different figures.

**Response:**

We agree with the referee's suggestions at this. The difference between figure 7(c) and 7(d) is that they were the averaged FLD from different time periods. The first time period is the same as VIPPHEN product (1981-2014), and the second time period is the same as VNP22C2 product (2013-2020). Figure 7(g) and 7(h) also show this difference.

**Changes in manuscript:**

We have revised Figure 7 by adding time period labels in each subfigure to make it clearer.

**(6) Reference problems:** a) line 491, please check this reference. b) please delete some not very relevant references, since there were some many references cited in the manuscript.

**Response:**

We appreciated your suggestions very much.

**Changes in manuscript:**

a) This reference was not complete. We have modified it to the correct format. ("Lieth, H.: Purposes of a phenology book, in: Phenology and Seasonality Modeling, edited by: Lieth, H., Springer, Berlin, Heidelberg, 3–19, https://doi.org/10.1007/978-3-642-51863-8_1, 1974. ")

b) We have rechecked all references in this study and removed eight less relevant references from the Introduction.

**(7)** Finally, please state the reason why you used species distribution probability as a weight when aggregated species phenology to community phenology. Please give a more detailed explanation in the text at this.

**Response**:

Thank you very much for your mention of the weights used in the aggregation method. In previous studies, species abundance was usually used as the weight of each species when aggregation phenology from species level to community level (Liang et al., 2011). However, this method is limited to the site scale, because species abundance data on the regional scale are extremely difficult to obtain. In this study, we innovatively proposed to use species distribution probability instead of species abundance as the weight. This assumption is based on the fact that species distribution and species abundance have a potentially positive relationship (Brown, 1984; Sagarin and Gaines, 2002).

**Changes in manuscript:**

We have added a more detailed explanation on Lines 186-190.

**Reviewer: 2**

**Comments from the Referee**

This study uses 24552 in-situ phenology observation records of 24 typical woody plants from the Chinese Phenology Observation Network, five species-level phenology models, weighted average, and quantile methods, to map the species and ground phenology of forests for 1951-2020 with 1-day temporal and 0.1° spatial resolution. The results show that the SP maps in this study have the same or smaller simulation error, and the GP maps in this study have good agreement with other LSP products, so the data has high accuracy and reliability. The methods in this paper are valid and feasible. The authors first provide the dataset of the phenology map of China, and this dataset is crucial in promoting our understanding of the spatial pattern of plant phenology along the geographic gradient, and the temporal trends of plant phenology across China, and providing support for global change impact assessment, terrestrial ecosystem simulation, and natural resource management. Overall, the topic is relevant and suited to the audience of ESSD, and I support publishing this manuscript in *Earth System Science Data* after some minor revisions.

**Minor comments:**

**L26-31**: Plant phenology is a highly sensitive biological indicator of climate change; for example, climate warming significantly advances plant spring phenology but the rising temperature induced a lower temperature sensitivity of spring phenology, likely due to a warmer winter and water availability, see Fu et al., 2015 in Nature

**Response:**

Thank you very much for the point that complemented the manuscript. Previous study has found that climate warming significantly advanced plant spring phenology but reduced their temperature sensitivity, and verified that plant phenology is a sensitive indicator of climate change (Fu et al., 2015).

**Changes in manuscript:**

We have referred to this article (Line 29), and added it to the reference list.

**L78-79**: The authors state that "*Three phenophases, namely the first leaf date (FLD), first flower date (FFD), and 100% leaf coloring date (LCD), were included for each species.*" It is no problem, and the results are okay based on these phenophases. I suggest that the authors try to analyze the data using the phenophases derived from different percentages (e.g., 20%, 50%, 80%) in the future, and maybe, a nice result could be found.

**Response:**

We appreciated your suggestions very much. In this study, we used only the three most accessible phenophases (FLD, FFD, LCD) that were well documented in the Chinese Phenology Observation Network (CPON). A total of 18 phenophases were recorded for each woody plant species in the observation of CPON. In addition to the three we considered, there are other phenophases derived from different percentages, such as 50% leaf unfolding date, 90% leaf unfolding date, and 10% leaf coloring date etc. We will consider using these phenophases for further analysis and look forward to obtaining

better results in the future studies.

**L95**: Please give more information about "*the principle of three sigma limits*".
**Response:**
Thank you very much for your mention of the "*the principle of three-sigma limits*". Three-sigma limits is a statistical calculation where the data are within three standard deviations from a mean (Pukelsheim, 1994). In this study, three-sigma limits were used to set the upper and lower control limits of phenology data. On a standard normal distribution curve, data beyond the three-sigma lines represents less than 1% of all data points and are therefore identified as outliers.
**Changes in manuscript:**
We've added more information in the revised version (Line 94-96).

**Figure 2 and the following**: For the "$R^2$", italicize "R" in the full text.
**Figure 5**: Replace "R^2" with "$R^2$" in all relevant figures.
**L251 and the following**: For the "R", replace "R" with "r" and italicize "r" in the full text.
**Response**:
Thank you very much for your reminder at these.
**Changes in manuscript:**
For the goodness of fit ($R^2$), we have changed "R^2" into "$R^2$" in Figure 5, and italicized "R" in the full text. For the correlation coefficient (r), we have changed "R" into lowercase "r", and italicized it in all relevant figures and text.

**L266 and following**: There are a lot of acronyms, so it is hard to follow as a generalist reader. I advise that the paper abbreviate these terms about the phenophases, and give the full name for the other terms, such as vegetation types.
**Response:**
We fully agree with the referee's suggestions at this.
**Changes in manuscript:**
We have modified the abbreviations for all vegetation types in the text to their full names, including EF (evergreen forests) MF (mixed forests), and DF (deciduous forests). In addition, we have changed some of the abbreviations "SP (species phenology)" into "phenology" or "phenology data" to make it clearer.

**L353-360:** The driving mechanisms for the autumn phenology are complex, for example, temperature has large effects on the autumn phenology, see Fu et al., "Larger temperature response of autumn leaf senescence than spring leaf-out phenology".
**Response:**
Thank you very much for providing this important reference. We have already cited this article's perspective on the mechanism of autumn phenology in the discussion section, and added it to the reference list.
**Changes in manuscript:**
The modification statement is as follows (Line 365-367): "*Moreover, the driving mechanisms for the autumn phenology are complex, which poses an additional challenge.*

*For example, temperature has large effects on the autumn phenology than the spring phenology (Fu et al., 2018)."*

**Related References appeared in the response letter**

Brown, J. H.: On the Relationship between Abundance and Distribution of Species, Am. Nat., 124, 255–279, https://doi.org/10.1086/284267, 1984.

Fu, Y. H., Zhao, H., Piao, S., Peaucelle, M., Peng, S., Zhou, G., Ciais, P., Huang, M., Menzel, A., Peñuelas, J., and others: Declining global warming effects on the phenology of spring leaf unfolding, Nature, 526, 104–107, https://doi.org/10.1038/nature15402, 2015.

Fu, Y. H., Piao, S., Delpierre, N., Hao, F., Hänninen, H., Liu, Y., Sun, W., Janssens, I. A., and Campioli, M.: Larger temperature response of autumn leaf senescence than spring leaf-out phenology, Global Change Biol., 24, 2159–2168, https://doi.org/10.1111/gcb.14021, 2018.

Liang, L., Schwartz, M. D., and Fei, S.: Validating satellite phenology through intensive ground observation and landscape scaling in a mixed seasonal forest, Remote Sens. Environ., 115, 143–157, https://doi.org/10.1016/j.rse.2010.08.013, 2011.

Pukelsheim, F.: The three sigma rule, Am. Stat., 48, 88–91, https://doi.org/10.2307/2684253, 1994.

Sagarin, R. D. and Gaines, S. D.: The 'abundant centre' distribution: to what extent is it a biogeographical rule?, Ecol. Lett., 5, 137–147, https://doi.org/10.1046/j.1461-0248.2002.00297.x, 2002.

---

## Author Response (AR2)

Dear Editor,

We are trying to respond to all the comments in details as below. We have revised the manuscript and addressed the points raised by the topic editor. We have improved the language and added a cover page to the supplemental material in this revision.

Many thanks for all the work all the editors have done on this manuscript. We hope this revision can solve most of the problems you referred to in your comments.

Please find the specifics below.

**Comments from the Editor**:

Dear authors,

Thanks for addressing the comments that reviewers raised. This work presents an informative and robust database;

(1) however, the manuscript would benefit from further language refinement.

(2) Additionally, please include a cover page for the supplementary material and ensure that the supplementary tables and figures are appropriately cited in the main text.

**Response:**

(1) We appreciated your suggestions very much. In the new version, we have made great efforts to polish and refine the language of the full text, improving clarity and overall readability, and making it more in line with academic standards.

(2) Thank you very much for your reminder at this. We have included a cover page in the supplementary material in accordance with the journal's format. We also double-checked all references to the supplementary material in the main text and found that Table S3 was missing from the previous version. In the new version, we have appropriately cited Table S3 in the main text (line 236).